# Scrutinizing the protein hydration shell from molecular dynamics simulations against consensus small-angle scattering data

Johanna-Barbara Linse [1] & Jochen S. Hub [1✉]

Biological macromolecules in solution are surrounded by a hydration shell, whose structure differs from the structure of bulk solvent. While the importance of the hydration shell for numerous biological functions is widely acknowledged, it remains unknown how the hydration shell is regulated by macromolecular shape and surface composition, mainly because a quantitative probe of the hydration shell structure has been missing. We show that small-angle scattering in solution using X-rays (SAXS) or neutrons (SANS) provide a protein-specific probe of the protein hydration shell that enables quantitative comparison with molecular simulations. Using explicit-solvent SAXS/SANS predictions, we derived the effect of the hydration shell on the radii of gyration $R_g$ of five proteins using 18 combinations of protein force field and water model. By comparing computed $R_g$ values from SAXS relative to SANS in $D_2O$ with consensus SAXS/SANS data from a recent worldwide community effort, we found that several but not all force fields yield a hydration shell contrast in remarkable agreement with experiments. The hydration shell contrast captured by $R_g$ values depends strongly on protein charge and geometric shape, thus providing a protein-specific footprint of protein–water interactions and a novel observable for scrutinizing atomistic hydration shell models against experimental data.

---

[1] Theoretical Physics and Center for Biophysics, Saarland University, Saarbrücken 66123, Germany. ✉email: jochen.hub@uni-saarland.de

Water molecules play key roles in protein functions such as folding, molecular recognition, enzymatic activity, and proton transfer[1–3]. During such functions, water interacts with the geometrically rough and chemically heterogeneous protein surface by the formation of hydrogen bonds with polar and ionic groups as well as by long-ranged Coulomb and Van-der-Waals forces. Protein–water interactions together with water-internal interactions lead to the formation of a water layer with different structural and dynamic properties as compared to bulk water, termed protein hydration shell. The modified water dynamics in the hydration shell have been studied by NMR and Terahertz spectroscopy, time-dependent fluorescence Stokes shift, inelastic neutron scattering, molecular dynamics (MD) simulations, and several other techniques[4–13]. These data revealed that geometric constraints and the hydrogen bond network lead to a mild slowdown of water dynamics by factors of 3–5 and to an increased water ordering as compared to bulk water[14]. The importance of protein–water interactions is further augmented in crowded cellular environments, where macromolecules typically adopt 25–40% of the volume[15]. In such environments, up to 70% of the water is part of a biomolecular hydration shell[3], demonstrating that biology largely involves non-bulk-like water[16].

Whereas the dynamics of the protein hydration shell has been investigated in great quantitative detail by spectroscopy, the overall structure and contrast of the hydration shell is far less understood. Small-angle scattering (SAS) with X-rays (SAXS) or neutrons (SANS) revealed that the hydration shell of many proteins exhibit an increased density compared to the bulk[17], which imposes an increased or a decreased radius of gyration $R_g$ as detected by SAXS or by SANS in $D_2O$, respectively. MD simulations with explicit solvent reproduced the modified $R_g$ values and attributed this effect to an excess density of ~6%[18,19]. A recent combined SAXS/SANS study on ultra-charged proteins suggested that anionic surface amino acids (Asp/Glu) enhance the hydration shell density more as compared to cationic amino acids (Arg/Lys)[20]. Because data from SAS or spectroscopy provide information on the hydration shell with only low spatial resolution or low information content, atomic insight into protein hydration mostly relies on MD simulations[16,18,21–23]. However, whether current protein force fields and water models accurately capture the structure of the protein hydration shell is not known.

Apart from the pioneering study by Svergun et al.[17], SAS data has been hardly used as a probe for protein hydration, mostly because undesired scattering contributions from protein aggregation, radiation damage, or poor buffer matching are hard to quantify with sufficient accuracy from a single SAS experiment, despite progress in quality standards for SAS experiments and analysis[24,25]. Thus, to obtain high-precision SAS data for benchmarking and to test the reproducibility of SAS data, a recent worldwide round-robin study collected SAS data for five globular proteins at 12 SAXS and four SANS instruments, leading to a total of 247 SAS curves[26]. The five proteins considered in the round-robin study were selected to be relatively rigid to avoid complications owing to pronounced flexibility. Whereas SAS data for lysozyme and urate oxidase were subject to increased uncertainties (Supplementary Discussion), SAS data for ribonuclease A (RNaseA), xylanase, and glucose isomerase were reproducible at many beamlines, thus providing consensus SAS data with unprecedented accuracy.

Here, we tested whether all-atom MD simulations reproduce the increased density of the protein hydration shell. To this end, we carried out simulations using 18 different combinations of protein force field and water model (Table S1), computed SAS curves taking explicit solvent into account[18,27–30], and compared the derived $R_g$ values to consensus SAS data from the round-robin study (Table S2)[26]. We included widely used variants of the CHARMM[31] and AMBER force fields families[32–35] and ten different water models, including several recently proposed force field combinations with increased protein–water dispersion interactions[35–38]. Overall, we find nearly quantitative agreement between simulation and experiments for many force field combinations; however, the calculations furthermore reveal deviations relative to experiments for certain protein force fields or water models. Thus, the quantitative comparison involving high-precision SAS data and explicit-solvent SAS calculations provides a novel route for scrutinizing the structure of the protein hydration shell.

## Results

**Explicit-solvent SAS calculations reveal the hydration shell effect on $R_g$.** The three-dimensional solvent density around xylanase is illustrated in Fig. 1b and in Supplementary Movie 1, computed from a simulation carried out with the ff99SBws protein force field and the TIP4P/2005s water model (Fig. S1)[36,39]. The density reveals the first hydration layer, which is structured by the formation of favorable interactions between the solvent molecules and the protein surface (red/orange mesh), as well as the second hydration layer (blue mesh), which is more dispersed. By averaging the solvent density over the protein surface, the solvent density is obtained as a function of the distance from the Van-der-Waals surface of xylanase, revealing, in addition to the pronounced first and second hydration layer a shallow third layer at a distance of ~7 Å (Fig. 1c, solid magenta line), as reported by many previous MD studies (Ref. [19] and references therein). To test whether solvent density modulations owing to water–protein interactions differ from density modulation owing to the internal structure of bulk water, we carried out a simulation of bulk TIP4P/2005s water, in which water molecules within a volume similar to the volume of xylanase were restrained (Supplementary Methods). Density modulations around such restrained bulk water are by far smaller as compared to the density modulations at the protein surface (Fig. 1c, compare dashed dark green with solid magenta line), in line with previous reports[18]. Thus, explicit-solvent MD simulations yield the structure of the hydration shell that differs from the structure of bulk solvent and, thereby, manifests as a modified radius of gyration $R_g$ detected by SAS experiments[17].

Using explicit-solvent SAS calculations[29,40], we computed from MD simulations SAXS curves, SANS curves in $H_2O$, and SANS curves in $D_2O$ as function of momentum transfer $q$, where $q = 4\pi \sin(\theta)/\lambda$ with the scattering angle $2\theta$ and the wavelength $\lambda$ of the X-ray beam (Fig. 1d). Two approaches may be used to extract $R_g$ from the SAS intensity curves $I(q)$: (i) via the the Guinier fit $\ln[I(q)/I_0] \approx -(qR_g)^2/3$ to the small-$q$ region, where $I_0$ is the forward scattering intensity (Fig. 1d, inset); or (ii) via the the pair distance distribution function (PDDF), also referred to as $P(r)$ function, which is obtained from the SAS curve via a regularized inverse Fourier transform[41,42], providing the radius of gyration via $R_g^2 = \int r^2 P(r)\,dr / [2 \int P(r)\,dr]$. From the simulations, we report $R_g$ obtained with the Guinier fit, yet we validated the agreement with the $R_g$ obtained from the PDDF. Both, $R_g$ and $I_0$ are influenced by the contrast of the hydration shell relative to the bulk solvent. However, because the experimental uncertainties of $R_g$ are by far smaller as compared to uncertainties of $I_0$, we validated MD simulations against experimental $R_g$ values in this study.

Because our SAXS and SANS calculations take explicit water molecules in the hydration shell into account, the $R_g$ and also $I_0$ values are fully controlled by the water and protein force fields (together with MD parameters such as cutoffs). We quantified the effect of the hydration shell on the $R_g$ by computing the difference

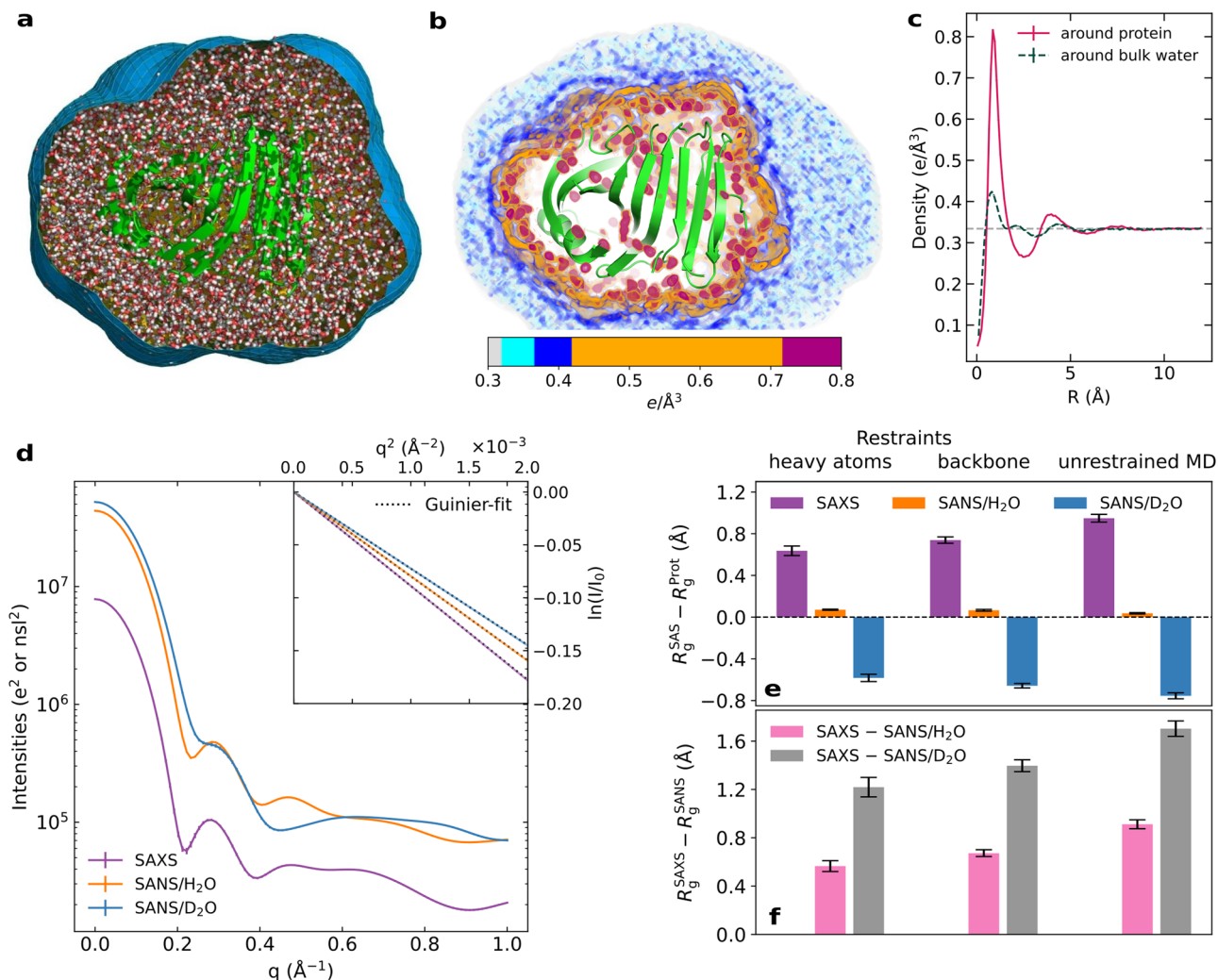

**Fig. 1 Explicit-solvent MD reveals the hydration shell structure and modified $R_g$ values from SAXS and SANS. a** Simulation of xylanase obtained with ff99SBws and TIP4P/2005s water. Water molecules within the envelope (blue surface) contribute to SAS calculations. Water outside of the envelope is not shown for clarity. The protein is shown in green cartoon, water as red/white sticks. **b** Electron density of solvent inside the envelope in shades from light gray (bulk water) to blue to orange to red, revealing the first (orange and red) and the second (mostly blue) hydration layers. **c** Solvent density versus distance $R$ from the Van-der-Waals surface of the protein, averaged over the protein surface (magenta solid line), revealing two pronounced and a third weak hydration shell. The solvent density around a volume of restrained bulk water (dark green dashed line) reveals by far smaller modulations, demonstrating that water–protein interactions lead to a more structured and more dense hydration shell compared to bulk water. The experimental bulk density of 0.334 $e/Å^3$ is shown by a gray dashed line. **d** Calculated intensity curves for SAXS (purple), SANS in $H_2O$ (orange), and SANS in $D_2O$ (blue) obtained from MD simulations. Curves are shown in absolute units of $e^2$ for SAXS and squared neutron scattering lengths (nsl$^2$) for SANS. Inset: Guinier plots of SAS curves (colored lines) and linear fits (dotted black lines) used to obtain the SAS-derived radii of gyration $R_g$. **e** Difference between SAS-derived $R_g$ values and the $R_g$ values of the pure protein ($R_g^{Prot}$) for SAXS, SANS/$H_2O$, and SANS/$D_2O$ (color code as in panel d). $R_g$ differences were computed from simulations with restrained heavy atoms (left), restrained backbone (middle), or from unrestrained MD (right). **f** Differences between $R_g$ from SAXS and SANS/$H_2O$ (pink), as well as from SAXS and SANS/$D_2O$ (gray). All $R_g$ differences are a footprint of the protein hydration shell. Statistical errors denote 1 SE.

$\Delta R_g = R_g^{SAS} - R_g^{Prot}$ between the $R_g$ from the SAS curve, $R_g^{SAS}$, and $R_g$ calculated from the atomic positions of protein atoms, $R_g^{Prot}$. The $\Delta R_g$ values calculated from simulations of xylanase with restraints on heavy atoms or on backbone atoms or from unrestrained MD simulations are shown in Fig. 1e, demonstrating that the hydration shell modulates $R_g$ of xylanase by up to 0.9 Å.

**Differences in $R_g$ values obtained from SAXS relative to SANS/$D_2O$ enables quantitative comparison between MD simulations and SAS experiments.** The effect of the hydration shell on $R_g$ is different in SAXS as compared to SANS experiments (Fig. S3). Because X-rays scatter at the electrons whereas neutrons scatter at

the nuclei, SAXS curves report on the electron density contrast, whereas SANS curves report on the contrast of the neutron scattering length density. Many globular proteins exhibit a hydration shell with an increased electron density relative to the bulk solvent[17,18,29]. For such proteins, both, the protein and the hydration shell exhibit a positive electron density contrast relative to the bulk (Fig. S3a), leading in a SAXS experiment to an increased $R_g$ ($\Delta R_g > 0$, Fig. 1e, purple bars). For SANS in $D_2O$, the protein exhibits a negative contrast of the neutron scattering length density whereas the hydration shell exhibits a positive contrast relative to bulk, resulting typically in a decreased $R_g$ ($\Delta R_g < 0$, Fig. S3b, Fig. 1e, blue bars). For SANS in $H_2O$, the contrast of the protein is positive whereas the contrast of the

hydration shell is close to zero, leading to a small influence by the hydration shell on $R_g$ ($\Delta R_g \approx 0$, Fig. S3c, Fig. 1e, orange bars).

Because the solution structure of proteins may differ from their crystal structure, $R_g^{Prot}$ present in SAS experiments is not accurately known. Hence, we focused here on the difference $\Delta R_g^{SAS} = R_g^{SAXS} - R_g^{SANS}$ between the $R_g$ values from SAXS ($R_g^{SAXS}$) and SANS ($R_g^{SANS}$). For the xylanase simulations discussed above, $\Delta R_g^{SAS}$ takes values up to 1.7 Å or 0.9 Å for SANS in $D_2O$ or $H_2O$, respectively (Fig. 1f, gray or pink bars, respectively). In contrast to $\Delta R_g$ values, $\Delta R_g^{SAS}$ values do not depend on $R_g^{Prot}$, thus enabling direct comparison between simulation and experiment.

**The hydration shell effect on $R_g$ increases upon solvent relaxation on a flexible protein surface.** We first evaluated the effect of atomic fluctuations on the hydration shell and, thereby, on $\Delta R_g$ and $\Delta R_g^{SAS}$. To this end, a series of simulations of xylanase was carried out with increasing flexibility by using restraints on heavy atoms, on backbone atoms, or by running an unrestrained MD simulation. Figure 1e, f demonstrates that the fewer atoms of a protein were restrained, the larger is the influence of the hydration shell on $\Delta R_g$, indicating a hydration shell with increasing contrast. To shed more light on the effect of protein flexibility on $\Delta R_g^{SAS}$, we carried out additional simulations of restrained xylanase after unrestrained equilibration. Notably, after such unrestrained equilibration, applying restraints to the protein had on average no effect on $\Delta R_g^{SAS}$ (Fig. S4b, c). Thus, protein flexibility takes effect on $\Delta R_g$ and $\Delta R_g^{SAS}$ during the equilibration phase, enabling more favorable relaxation of water onto the protein surface as compared to simulations with restraints. With increasing conformational flexibility, the water molecules may favorably pack on the protein surface and penetrate cavities between side chains, which leads to an increasingly dense hydration shell. Below, we use results from unrestrained MD simulations for force field validations against experimental data.

**Comparison of the hydration shell from 18 force field combinations with consensus SAS data.** Next, we studied the effect of 18 different combinations of force fields for protein and water on the hydration shell, as quantified by $\Delta R_g$ and $\Delta R_g^{SAS}$ values. We considered widely used force field combinations such as CHARMM36m–TIP3P[31,43] as well as uncommon combinations such as CHARMM36m–SPC/E[31,44] (Table S1). Such uncommon combinations are generally not recommended because protein force fields have been parametrized with respect to specific water models; in this study, however, we considered such uncommon combinations with the aim to dissect effects of the protein force field and of the water model on the hydration shell. Figure 2a–c presents $\Delta R_g$ values from unrestrained MD simulations, i.e., the difference of $R_g$ values from the Guinier analysis relative to $R_g^{Prot}$ values. Absolute $R_g$ values are shown in Fig. S5 and all computed $R_g$ values are listed in Tables S3–S7. Among all force fields, $\Delta R_g$ values were positive, near-zero, and negative for SAXS, SANS/$H_2O$, and SANS/$D_2O$, respectively, in line with results in Fig. 1e. However, $\Delta R_g$ values vary considerably among different force fields by up to 0.5 Å, indicating different contrasts in the hydration shell. For instance, ff99SBws–TIP4P/2005s, which implements increased water–protein dispersion with the aim to obtain realistic ensembles of intrinsically disordered proteins[36,39], yields by far larger modulations of $R_g$ (black bars) as compared to CHARMM36m with the CHARMM-modified TIP3P water model (cTIP3P, blue bars) or with the OPC water model[31,43,45]. These differences propagate into differences of $\Delta R_g^{SAS}$,

which we compare with consensus experimental SAS data in the following.

Figure 3b, d, f presents $\Delta R_g^{SAS}$ values obtained from SAXS relative to SANS/$D_2O$ for the proteins RNaseA, xylanase, and glucose isomerase, computed with 18 different combinations of force fields for protein and water and using unrestrained simulations. For many force fields, excellent agreement is found between simulation and experiment (Fig. 3, horizontal lines), suggesting that many force fields yield a correct overall hydration shell contrast. Such agreement is remarkable considering that protein–water interaction potentials have not been refined against solution scattering data but rather against thermodynamic data such as hydration free energies[46]. However, Fig. 3b, d, f furthermore reveals considerable differences among protein force fields and water models. These $\Delta R_g^{SAS}$ differences reveal similar trends if derived from backbone-restrained instead of from unrestrained simulations (Fig. S6), demonstrating that $\Delta R_g^{SAS}$ variations among force fields are mostly not a consequence of different protein conformations but instead a consequence of different packing of water on the protein surface.

To test whether $\Delta R_g^{SAS}$ variations among force fields are explained by variations of the hydration shell densities, we computed solvent density profiles as function of distance $R$ from the Van-der-Waals surface of xylanase using the force field combinations CHARMM36m–cTIP3P, ff15fb–TIP3P-FB, or ff99SBws–TIP4P/2005s (Fig. S7), which led to small, medium, or large $\Delta R_g^{SAS}$ values, respectively (Fig. 3d, blue, green, and black bar, respectively). In addition, density profiles around xylanase were compared with density profiles around volumes of restrained bulk water modeled with cTIP3P, TIP3P-FB, or TIP4P/2005s (Fig. S8), thereby comparing the hydration shell structures with the structure of bulk water[18]. We find that the height and width of the first solvent density peak at $R \approx 0.85$ Å correlate with $\Delta R_g^{SAS}$ values, both if taken from the total solvent density profiles (Fig. S7, S8) or if taken from the density profile relative to the bulk water structure (Fig. S8b, d). This analysis suggests that modulations of the hydration shell structure among different force fields indeed manifest in variations of $\Delta R_g^{SAS}$.

**Force field effects on the hydration shell are different for anionic compared to near-neutral proteins.** Focusing first on the weakly charged proteins RNaseA and xylanase, $\Delta R_g^{SAS}$ values obtained with CHARMM36m are systematically lower relative to the experiment, irrespective of the applied water model SPC/E, TIP3P, or OPC3 (Fig. 3b, d). An exception is given by the RNaseA simulation with OPC, for which CHARMM36m led to a larger $\Delta R_g^{SAS}$ as compared to ff99SBws. However, this difference is inverted in simulations with backbone restraints (Fig. S6a, b), suggesting that the simulation with CHARMM36m–OPC adopted an unusual conformation, as confirmed by visual inspection of the trajectory (Fig. S9). Thus, in conformationally stable simulations, CHARMM36m imposed a less dense hydration shell as compared to the tested AMBER force field variants. Among all tested force fields, CHARMM36m–cTIP3P yields the lowest $\Delta R_g^{SAS}$ values, indicating a hydration shell with too low contrast. The AMBER force field variants ff14SB and ff99SB-ildn show good agreement with the experiment with any of the water models SPC/E, TIP3P, OPC3, TIP4P-D, or TIP4P/2005. Hence, surprisingly, among the tested force field combinations, the protein force field has a larger effect on the hydration shell contrast as compared to the water model.

Several older force fields are not suitable for simulating intrinsically disordered proteins (IDPs) as they impose overly

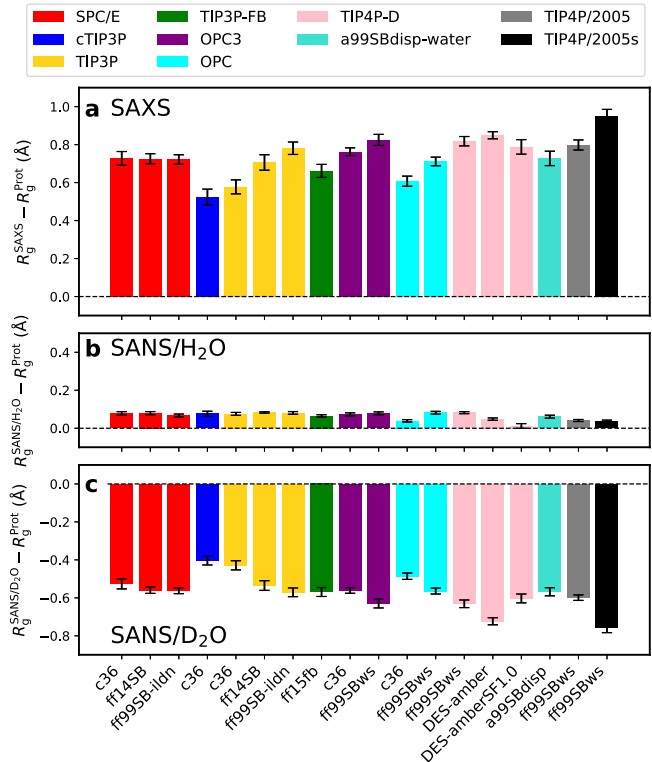

**Fig. 2 SAS-derived $R_g$ values of xylanase relative to $R_g$ of the bare protein.** Modulations $\Delta R_g$ of the SAS-derived $R_g$ values relative to $R_g$ of the bare protein from unrestrained simulations of xylanase, obtained with 18 different combinations of protein force field (labels along the abscissa) and water model (color code, see legend). **a** $\Delta R_g$ from SAXS, **b** from SANS in $H_2O$, and **c** from SANS in $D_2O$. Statistical errors (1 SE) were obtained from block averaging. For force field abbreviations, see Table S1.

collapsed IDP ensembles, which has been attributed to a lack of protein–water dispersion interactions[36,47]. Thus, several modifications of the AMBER force field family have been developed with refined protein–water dispersion interactions, leading to more expanded IDP ensembles. We considered the protein force fields DES-amber[35,37], DES-amber without scaled charges DES-amberSF1.0, a99SBdisp[38], and ff99SBws[36], each simulated in conjunction with the recommended water model TIP4P-D, a99SBdisp, or TIP4P/2005s (Table S1). For RNaseA and xylanase, the $\Delta R_g^{SAS}$ values of DES-amber and a99SBdisp are in good agreement with the experimental values, although smaller variations are visible (Fig. 3b, d). The combination of ff99SBws with TIP4P/2005s leads to large $\Delta R_g^{SAS}$ values, exceeding the experimental values. This may indicate that ff99SBws–TIP4P/2005s overestimates protein–water interactions, leading to overly dense hydration shells of RNaseA and xylanase (Fig. 3b, d, black bar).

Glucose isomerase is a globular protein decorated with a high density of acidic amino acids, leading to a large negative charge of $-60\,e$. Glucose isomerase exhibits large $\Delta R_g^{SAS}$ values of SAXS relative to SANS/$D_2O$, indicative of an exceptionally pronounced hydration shell (Figs. 3f and S6f). These findings agree qualitatively with Kim et al. who reported a pronounced hydration shell around a highly anionic variant of green fluorescent protein (GFP) but not around the wild-type or around a highly cationic variant of GFP[20]. All 18 force field combinations captured the increased $\Delta R_g^{SAS}$ values of glucose isomerase relative to RNaseA or xylanase. Many force field

combinations yield excellent agreement with the experimental value, however, several combinations even seem to overestimate $\Delta R_g^{SAS}$, indicative of a slightly exaggerated hydration shell contrast. Interestingly, variations among different force fields do not follow the same trend for glucose isomerase as observed for the near-neutral proteins RNaseA or xylanase; for instance, CHARMM36m simulations yield similar (with TIP3P) or even larger (with SPC/E or OPC3) $\Delta R_g^{SAS}$ values as compared to the simulations with the AMBER force fields ff14SB or ff99SB-ildn. Thus, force field effects on the hydration shell depend on the physicochemical characteristics of the protein surface, suggesting that comparisons based on several proteins are mandatory to scrutinize force field effects on the protein hydration shell.

SANS collected in $H_2O$ is subject to poorer signal-to-noise ratio as compared to SANS in $D_2O$ owing to the greatly increased incoherent scattering cross section of hydrogen relative to deuterium, leading to a pronounced scattering background. These challenges may lead to an inaccurate buffer subtraction and are reflected by largely increased uncertainties of the $\Delta R_g^{SAS}$ values obtained from consensus SAXS and SANS/$H_2O$ curves (Fig. 3a, c, e, horizontal orange lines, Table S2)[26]. Thus, current SANS/$H_2O$ data are not suitable for quantitative validation of the hydration shell from MD simulations. Instead, we suggest that MD simulations, after having validated the hydration shell against SAXS and SANS/$D_2O$ data as done here (Fig. 3b, d, f), may be used in future studies to improve the accuracy of the buffer subtraction of SANS/$H_2O$ experiments, thereby enabling a more quantitative analysis of SANS/$H_2O$ curves.

**Protein size and shape furthermore influence $\Delta R_g^{SAS}$ values.** Apart from SAS data of RNaseA, xylanase, and glucose isomerase, the round-robin SAS benchmark study collected SAS data for lysozyme and urate oxidase[26]. However, because the $R_g$ values of lysozyme and urate oxidase were subject to increased spread owing to problems with radiation damage and aggregation (SI Discussion), these data have not been used for validating MD simulations in this study. Instead, SAS calculations for lysozyme and urate oxidate provide additional insight on the effect of protein size and shape on $\Delta R_g^{SAS}$ values. MD simulations with 18 different force field combinations show by far larger $\Delta R_g^{SAS}$ values for lysozyme as compared to urate oxidase (Fig. S10b, d). We explain the large effect of the hydration shell on $R_g$ of lysozyme with the small size of protein (14.3 kDa), leading to a relatively large contribution of the hydration shell to the overall contrast of the solute. Urate oxidase (i) is larger (136.3 kDa) than lysozyme, leading to a smaller contribution by the hydration shell to the overall contrast and (ii) exhibits a solvent-filled cavity; because the hydration shell in the cavity adds contrast close to the center of mass of urate oxidase, part of the hydration shell may even decrease the $R_g$. Thus, $\Delta R_g^{SAS}$ values are not only controlled by the surface properties of the protein as emphasized by the case of glucose isomerase, but also strongly influenced by the size and geometric shape of the protein[48].

Figure 4 summarizes $\Delta R_g$ and $\Delta R_g^{SAS}$ values for five proteins, obtained as a consensus average over six combinations of protein force field and water model (Methods), which exhibited good agreement with experimental data according to Fig. 3b, d, f. As discussed above, Fig. 4 highlights that $\Delta R_g$ and $\Delta R_g^{SAS}$ values strongly depends on the protein and are rationalized by varying surface composition, protein size, and protein shape. Thus, $\Delta R_g^{SAS}$ values report a footprint of the hydration shell that reflects protein-specific protein–water interactions and enable quantitative comparison with the experiment.

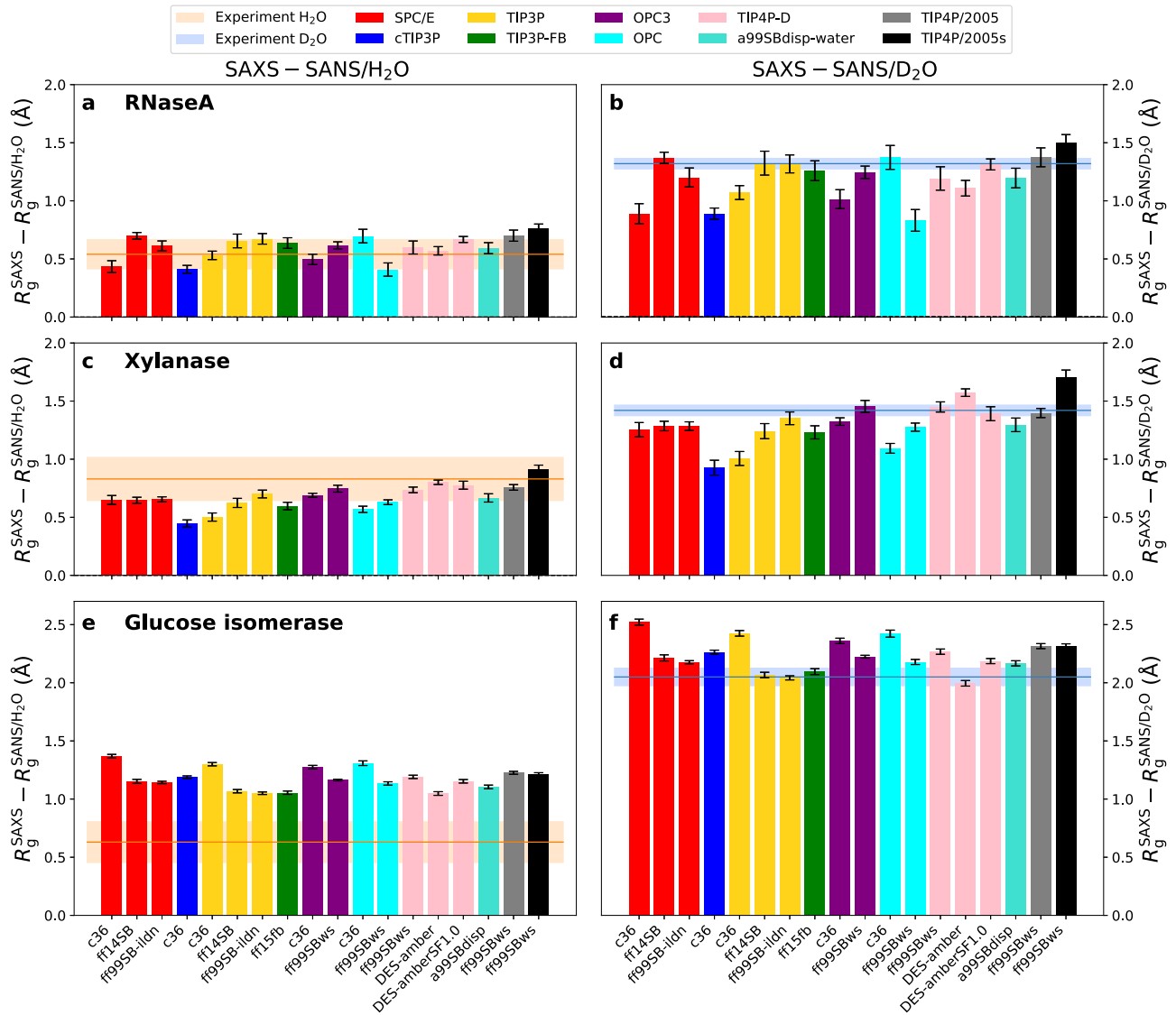

**Fig. 3 Difference in $R_g$ values from SAXS relative to SANS in $H_2O$ or $D_2O$.** Difference between $R_g$ values from SAXS and SANS/$H_2O$ (left column) or between SAXS and SANS/$D_2O$ (right column) obtained from unrestrained simulations of **a, b** RNaseA, **c, d** xylanase, and **e, f** glucose isomerase. $R_g$ values were obtained with 18 different combinations of protein force fields (labels along the abscissa) and water models (color code, see legend). Experimental consensus values and uncertainties from $P(r)$ analysis are shown as horizontal lines and shaded areas, respectively. Statistical errors denote 1 SE.

## Discussion

MD simulations have been widely used to study the structure and dynamics of the protein hydration shell or to study proteins in crowded environments, where only a minor fraction of the solvent is bulk-like[19,49–51]. Whether the solvent structure in the hydration shell and in crowded environments matches with experimental conditions remains unclear. We showed that consensus SAS data combined with explicit-solvent SAS calculations provide a novel means to validate the hydration shell from MD simulations. To this end, we focused on the difference $\Delta R_g^{SAS}$ between the $R_g$ values obtained with SAXS and SANS in $D_2O$, which (i) provides a measure for the contrast of the hydration shell relative to bulk solvent while it (ii) does not require accurate knowledge of the $R_g$ of the bare protein.

Overall, we found remarkable agreement between $\Delta R_g^{SAS}$ from MD simulations and experiments for many combinations of protein force field and water model for the proteins RNaseA, xylanase, and glucose isomerase. These results demonstrate that the hydration shell contrast and, thereby, the packing of solvent

on the protein surface is accurately captured by many modern force fields. However, we furthermore observed systematic differences among force field families. For the electrically nearly neutral proteins xylanase and RNaseA, simulations with CHARMM36m typically underestimated $\Delta R_g^{SAS}$ as compared to experimental values, in particular together with the widely used TIP3P or cTIP3P water models, indicative of underestimated hydration shell contrasts. Simulations with several AMBER variants revealed reasonable or even excellent agreement with experimental values. The $\Delta R_g^{SAS}$ values obtained with AMBER99SBws–TIP4P/2005s were larger compared to most other force fields and exceeded experimental values for several proteins, indicating that this force field yield a hydration shell with high contrast.

For the highly anionic glucose isomerase, and in contrast to simulations with near-neutral proteins, simulations with CHARMM36m revealed larger $\Delta R_g^{SAS}$ values as compared to most simulations with AMBER variants. These findings may suggest that CHARMM36m imposes tighter water packing on acidic

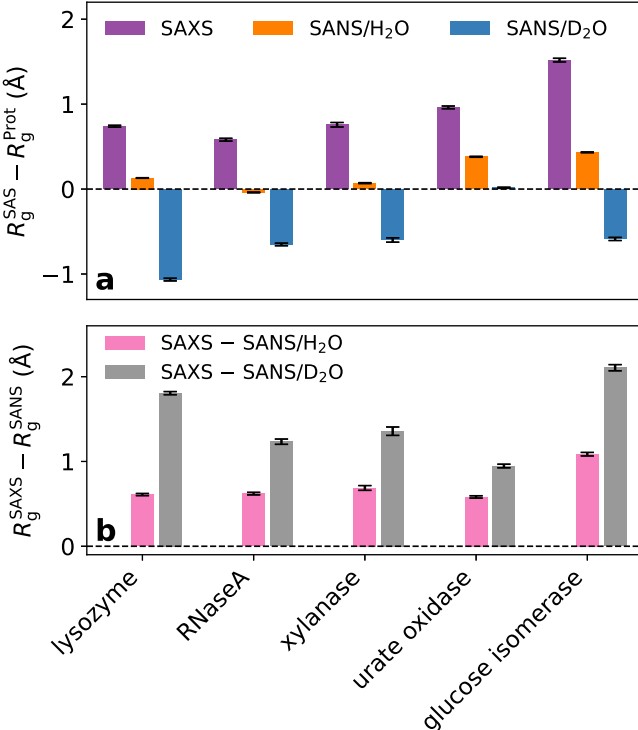

**Fig. 4 Computational consensus $\Delta R_g$ values and $\Delta R_g^{SAS}$ values for five proteins. a** Computational consensus $\Delta R_g$ values and **b** $\Delta R_g^{SAS}$ values for five proteins (see labels) obtained as average over six combinations of protein force field and water model that showed close agreement with experimental data according to Fig. 3. Color code is chosen following Fig. 1e, f.

residues as compared to AMBER force fields. However, we furthermore found that different coordination of ions[52] on acidic residues may contribute to variations of $\Delta R_g^{SAS}$ of glucose isomerase (Fig. S16).

Remarkably, the systematically different $\Delta R_g^{SAS}$ values with CHARMM36m relative to several AMBER variants were found irrespective of the applied water model, despite greatly different bulk properties of water models[53–55]. For instance, accurate bulk properties of a water model (such as OPC[45]) do not imply an accurate hydration shell contrast (if used together with CHARMM36m). Inversely, a water model with poor agreement with several bulk properties (such as TIP3P[45]) may yield an accurate hydration shell contrast (if used with ff14SB or ff99SB-ildn). Thus, $\Delta R_g^{SAS}$ yields structural information independent of the bulk properties that have been used to parametrize the water models. Protein–water interactions in MD simulations have frequently been validated using hydration free energies $\Delta G_{hyd}$ of amino acid analogues. However, $\Delta G_{hyd}$ values agree reasonably between CHARMM36m and recent AMBER variants[56], suggesting that $\Delta R_g^{SAS}$ provides a measure for protein–water interactions independent of $\Delta G_{hyd}$. In addition, protein–water interactions have been discussed in the context of IDP simulations. Simulations with AMBER–TIP3P yield overly compact IDP ensembles, which has been attributed to a lack of protein–water dispersion interactions[36,37,47], whereas CHARMM36m–TIP3P[31] or AMBER variants with increased dispersion interactions[37] yields IDP ensembles in better agreement with experiments. Thus, there exists no simple correlation between $\Delta R_g^{SAS}$ and the spatial extent of IDP ensembles. Together, these observations suggest that $\Delta R_g^{SAS}$ values represent a footprint of protein–water interactions that is independent of previously considered observables, thus

providing an additional observable for validating and further improving protein–water interactions in MD simulations.

Irrespective of the applied force field, $\Delta R_g^{SAS}$ values differed considerably between different proteins, in agreement with the data of the round-robin SAS study[26]. The highly anionic glucose isomerase exhibited the largest $\Delta R_g^{SAS}$ values among the five proteins considered in this study, indicative of a tightly packed hydration shell. These findings are in line with a SAS study of a highly anionic GFP variant[20] and demonstrate that the anionic aspartate and glutamate residues impose a densely packed hydration shell. Among the four proteins with zero or with a small net charge, lysozyme exhibited larger $\Delta R_g^{SAS}$ values as compared to urate oxidase, xylanase, and RNaseA (Fig. 4b). Urate oxidase exhibits the shape of a hollow cylinder with a large solvent-filled cavity, which may explain the low $\Delta R_g^{SAS}$ values (Fig. 4b) as well as a nearly vanishing $\Delta R_g$ for SANS/D$_2$O, in contrast to all other proteins (Fig. 4a). Thus, variations of $\Delta R_g^{SAS}$ are experimentally accessible footprints of protein-specific hydration shells reflecting specific geometric shapes or distributions of charged and polar moieties on the protein surface.

Because the $\Delta R_g^{SAS}$ values are in the range of only 1–2.5 Å, the comparisons presented here require highly accurate SAS data. Considering that SAS data may be subject to minor systematic errors, which may be difficult to detect, SAS data obtained at a single instrument may not yield the required accuracy, even if data collection and analysis follows established quality controls[24,25]. Instead, the use of consensus data collected at different SAS instruments, if possible, by independent researchers[26], is a rigorous means for obtaining data with unprecedented accuracy and, thereby, enables quantitative validation of the hydration layer as shown here. To validate the hydration shell of other biomolecules such as RNA or protein/RNA complexes, future benchmark studies similar to the round-robin study designed by Trewhella, Vachette, and coworkers would be of utmost value[26].

To enable quantitative comparison with the experiments, the MD simulations should match the experimental conditions and require control calculations. We carefully evaluated the effects of (i) protein flexibility (Figs. 1e, f and S4), (ii) use of salt as compared to use of only counter ions (Figs. S12–S15), (iii) refined sodium–carboxylate interaction parameters (Fig. S16) and (iv) Lennard-Jones cutoff settings (Fig. S17, see Supplementary Results). We found that these factors modulate $\Delta R_g^{SAS}$ only by a small fraction of an Ångström. Nevertheless, because such effects are clearly detectable in explicit-solvent SAS predictions of $\Delta R_g^{SAS}$, they require consideration upon comparison with experiments.

## Conclusions

We showed that the hydration shell contrast, as reported by SAS-derived $R_g$ values, strongly depends on the geometric shape and surface composition of proteins, thus providing a probe of protein-specific protein–solvent interactions. As readout of the hydration shell structure, we focused on the difference in $R_g$ values from SAXS relative to SANS experiments in D$_2$O ($\Delta R_g^{SAS}$), which we computed from MD simulations with explicit-solvent SAS calculations to enable quantitative comparison with experimental SAS data. For many force fields, $\Delta R_g^{SAS}$ values from MD simulations revealed excellent agreement with consensus SAS data from a recent world-wide round-robin study[26], suggesting that simulations accurately capture the hydration shell contrast relative to the bulk. Because we furthermore observed differences among force fields, our calculations provide the basis for further improving the accuracy of protein–water interactions in molecular simulations. This study establishes the combination of high-precision SAS experiments with explicit-solvent calculations as a tool for scrutinizing atomistic models of the protein hydration shell.

## Methods

**Simulation setup and parameters.** Initial structures of lysozyme, RNaseA, xylanase, glucose isomerase, and urate oxidase were taken from the protein data bank (PDB codes: 2VB1[57], 7RSA[58], 2DFC[59], 1MNZ[60] and 3L8W[61], respectively). Crystal waters were kept in the structures of lysozyme, RNaseA, and xylanase, whereas organic molecules of the crystallization buffer were removed. Missing amino acids of glucose isomerase and urate oxidase were added, such that the sequences were identical to the samples used in Ref. [26]. Specifically, one methionine was added to the N-terminus of glucose isomerase, and six amino acids (sequence (SLKSKL)) to the C-terminus of urate oxidase. Hydrogen atoms were added with the GROMACS module pdb2gmx. The starting structures were placed in a dodecahedral box, where the distance between the protein and the box edges was at least 2.0 nm, and the box was filled with explicit solvent. For the highly charged glucose isomerase ($-60\,e$), sodium and chloride ions were added by replacing the appropriate number of water molecules with ions to obtain a salt concentration of 100 mM NaCl. For all other proteins, the system was neutralized by adding sodium or chloride counter ions, if not stated otherwise.

Each protein was simulated using 18 combinations of protein force field and water model (Table S1). Interactions of the proteins were described with one of the following force fields: AMBER14SB (ff14SB)[32], AMBER99SB-ildn (ff99SB-ildn)[33], AMBER99SBws (ff99SBws)[39], AMBER99sb (ff99SB)[62], AMBER15/force-balance (ff15fb)[34], DES-amber[35], DES-amberSF1.0[35], AMBER99SBdisp (a99SBdisp)[38], or CHARMM36m (c36)[31] (version July 2020). Water was described with one of the following models: SPC/E[44], CHARMM-modified TIP3P involving hydrogen atoms with Lennard-Jones interactions (cTIP3P)[43], TIP3P[63], TIP3P-FB[64], OPC3[65], OPC[45], TIP4P-D[37], a99SBdisp-water[38], TIP4P/2005[66], or TIP4P/2005s[36].

All MD simulations were carried out with the GROMACS software, version 2020.3[67]. After 400 steps of minimization with the steepest decent algorithm, the systems were equilibrated for 100 ps with harmonic position restraints applied to the heavy atoms of the proteins (force constant 1000 KJ mol$^{-1}$nm$^{-2}$). Subsequently, the production runs were started without restraints on the atoms or with restraints applied to the heavy atoms (force constant 2000 KJ mol$^{-1}$nm$^{-2}$) or applied to the backbone atoms (force constant 2000 KJ mol$^{-1}$nm$^{-2}$) of the protein. The equations of motion were integrated using a leap-frog algorithm[68]. The temperature was controlled at 298.15 K, using velocity rescaling ($\tau = 1$ ps)[69]. The pressure was controlled at 1 bar with the Berendsen thermostat ($\tau = 1$ ps)[70] and with the Parrinello-Rahman thermostat ($\tau = 5$ ps)[71] during equilibration and production simulation, respectively. The geometry of the water molecules was constrained with the SETTLE algorithm[72] and LINCS[73] was used to constrain all other bond length. A time step of 2 fs was used. Dispersive interactions and short-range repulsion were described by a Lennard-Jones potential. For simulations with AMBER variants, LJ interactions were cut off at 1 nm. For simulations with CHARMM36m, LJ forces were gradually switched off between 1 nm and 1.2 nm, if not stated otherwise. In simulations with AMBER variants, the pressure and energy were corrected of missing dispersion corrections beyond the cut-off. Neighbor lists were updated with the Verlet scheme. Coulomb interactions were computed with the smooth particle-mesh Ewald method[74,75]. We used a Fourier spacing of ~0.12 nm, which was optimized by the GROMACS mdrun module at the beginning of each simulation. Systems with restraints on heavy atoms or on the backbone were simulated for 50 ns, which is sufficient to sample fluctuations of water molecules or amino acid side chains (if present). Unrestrained simulations were carried out for 230 ns. Because the five proteins considered in the round-

robin SAS study are relatively rigid, 230 ns were likely sufficient to sample the most relevant conformation space used to compute the increase of $R_g$ owing to the hydration shell.

The 3D solvent density shown in Fig. 1b was computed with the rerun functionality of GROMACS-SWAXS using the environment variable GMX_WAXS_GRID_DENSITY=1 and GMX_WAXS_GRID_DENSITY_MODE=2[76]. The solvent density written in CUBE format was visualized with PyMol[77].

**Explicit-solvent SAS calculations.** The SAXS and SANS calculations were performed with GROMACS-SWAXS (version 2021.5), a modified version of the GROMACS simulation software that implements explicit-solvent SAXS[29] and SANS calculations[78]. GROMACS-SWAXS is furthermore used by the web server WAXSiS for automated explicit-solvent SAXS predictions[30] and is freely available at GitLab (https://gitlab.com/cbjh/gromacs-swaxs). For more details on the rationale behind explicit-solvent SAS calculations including differences relative to implicit-solvent SAS calculations, we refer to previous reviews[76,79]. A spatial envelope (Fig. 1a) was constructed at a distance of 9 Å from all protein atoms. Solvent atoms (water and ions) inside the envelope contributed to the calculated SAXS/SANS curves, thereby taking the hydration shell into account. The buffer subtraction was carried out using 2251 simulations frames of pure solvent simulation box, which was simulated for 50 ns and large enough to enclose the envelope. The orientational average was carried out using 200 $\mathbf{q}$-vectors for each absolute value of $q$, and the solvent electron density was corrected to the experimental water density of 334 e/nm$^3$, as described previously[29]. In this study, a small number of only 200 $\mathbf{q}$-vectors per absolute value of $q$ was sufficient because we computed the SAS curves only up to small angles to carry out the Guinier analysis. The density correction is required to ensure accurate buffer matching of bulk solvent between the protein and the pure-water simulations, and to correct for the density of certain water models that differs from the experimental density. No fitting parameters owing to the hydration layer or excluded solvent were used, implying that the radius of gyration $R_g$ was not adjusted by the fitting parameters but fully imposed by the force field (together with other MD parameters such as cutoffs, temperature, etc.).

SAXS and SANS curves were computed from 2251 simulation frames taken from the time interval between 5 ns and 50 ns or between 30 ns and 230 ns for restrained and unrestrained simulations, respectively. Thus, computed SAS curves and $R_g$ values represent averages over protein and solvent fluctuations within the simulated time scales. Statistical errors of calculated SAS curves were obtained by binning the trajectories into 10 time blocks of 4.5 ns or 20 ns for simulations with or without restraints, respectively. Here, for restrained simulations, shorter simulations and time blocks used because restrained simulations exhibit shorter autocorrelation times as compared to unrestrained simulations. Likewise, the pure-solvent simulations were binned into independent blocks of 4.5 ns. Then the SAS curves were computed from independent pairs of solute and pure-solvent trajectories. Critically, the use of independent pure-solvent trajectory blocks is mandatory to exclude correlations between the SAS curves computed from time blocks. Reported error bars denote one standard error (1 SE).

SAXS data reported by the round-robin benchmark revealed only a marginal effect (if any) upon replacing H$_2$O with D$_2$O in SAXS experiments[26]. Thus, for the prediction of SANS/D$_2$O curves, we did not use force fields for heavy water[80] or force fields that would account for the deuteration of amino acids. Instead, we assigned the neutron scattering length of deuterium to water hydrogen atoms and to polar protein atoms. Hydrogen atoms of

the amine backbone groups were assumed to be deuterated with a probability of 90%.

To test whether computed $R_g$ values from Guinier analysis agree with $R_g$ values from the $P(r)$ function, we computed one SAXS curve of xylanase up to $q = 3\,\text{nm}^{-1}$, obtained the $P(r)$ with GNOM[81], and computed $R_g$ from $P(r)$. The $R_g$ values from Guinier and $P(r)$ analysis were identical and equaled 1.59 nm. Thus, we used computed $R_g$ from Guinier analysis for the remainder of this study for simplicity.

Computational consensus $R_g$ values were computed with the following combinations of protein and water force fields: ff14SB–TIP3P, ff99SB-ildn–TIP3P, ff15fb–TIP3P-FB, ff99SB ws–TIP4P-D, DES-amber–TIP4P-D, a99SBdisp–a99SBdisp-water.

Density profiles around proteins (Fig. 1c, solid line; Fig. S7) were computed from heavy atom-restrained simulations using gmx genenv module of GROMACS-SWAXS, by using an icosphere with 81920 triangular faces (option -nrec 6). The tool builds a series of envelopes with increasing distances from the Van-der-Waals surface of the protein and computes the electron density between pairs of adjacent envelopes[82]. Simulations used for computing density profiles around bulk water were set up as described in the Supplementary Methods (Fig. 1c, dashed line; Fig. S8).

### Data availability

All data used in this manuscript are available from the authors. MD input files to reproduce this study are publicly available in a Zenodo repository at https://doi.org/10.5281/zenodo.10007526[83].

### Code availability

MD simulation were carried out with GROMACS version 2020.3, which is freely available at https://www.gromacs.org. Explicit-solvent SAS calculations and density calculations were carried with GROMACS-SWAXS version 2021.5, which is freely available at https://gitlab.com/cbjh/gromacs-swaxs and documented at https://cbjh.gitlab.io/gromacs-swaxs-docs.

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

## Acknowledgements

We thank Jill Trewhella for insightful discussions. This study was supported by the Deutsche Forschungsgemeinschaft (DFG, German Research Foundation) via grants HU 1971/3-1, HU 1971/3-2, and INST 256/539-1.

## Author contributions

J.-B.L. performed and analyzed MD simulations. J.-B.L. and J.S.H. designed the project and wrote the manuscript.

## Funding

## Competing interests

The authors declare no competing interests.
