## [Peer Review File · Communications Chemistry]

Reviewers' comments:

Reviewer #1 (Remarks to the Author):

This is an interesting paper that compares several force-fields and water models to consensus SAXS and SANS data. The paper is well-written and provides an important contribution for the interpretation of SAXS/SANS data and for the assessment of force-field accuracy against scattering data.

My only comment on this work is that a quantity that has been suggested to be very important, the excess water density at the protein surface, is mentioned but never computed in this work. I recommend that the authors to compute this quantity (see e.g. figure 3 of ref. 18) and report how much it varies between force-fields and how much of this variation can explain variations in the results obtained here for different proteins/force-fields.

Reviewer #2 (Remarks to the Author):

In this manuscript by Linse et. al., the authors examine the effect of molecular dynamics force fields on the simulated hydration shells of five proteins by comparison against small angle scattering data (x-ray and neutron). They found that the hydration shell is dependent on the shape and the charge of the protein and is accurately described by only a few force-field combinations. Overall, the manuscript presents a meticulous detail of radius of gyration calculations for several systems while describing the best practices for comparing simulations with SAS experimental data and provides a very efficient dataset for further improvement of the accuracy of molecular simulations. The work will be a great resource for the molecular modeling community when published.

Addressing the following minor comment will improve the manuscript.

1. For Figure 1, it would be beneficial to have an image of the system with the complete water box for visual clarity, to show how much of bulk water is present in the MD simulation outside the hydration shells as shown in panels A and B.
2. The authors describe the presence of three hydration shells in xylanase, Fig 1, using solvent density averaged over the protein surface. However, there is no section in the Methods that describes this averaging procedure, detailing how the surface was determined for carrying out this analysis.
3. Figure S3 or its variant should be moved to the main text. The RMSD is the most widely used metrics of protein force field and hence will be of direct interest to the readers.
4. The time blocks selected for averaging ΔR_g values and calculating standard errors are different for the restrained and free simulations (4.5 ns vs 20 ns) for the same analysis. Were the pure solvent simulations binned into the same time blocks? It is not very clear why they have selected different time blocks for ensuring uncorrelated data?

5. The authors have done a throughout job of detailing all the average Rg data for the simulations carried out in main text figures and SI figures and tables. To supplement this, it would be useful to add the individual traces of RgProt, to check if there are unusual structural deformations that lie outside the calculated average values, similar to the RMSD data shown in Fig. S3A. It should also help to single out the unusual conformations in the CHARMM36m-OPC simulation that the authors report in Fig. S6 from visual inspection of the corresponding trajectory.

6. Following onto the previous comment, it would also useful to visualize the protein conformations that show the best agreement with experimental SAS data. A representative protein conformation along with the hydration shell can be shown for the best performing force-field combinations shown in Figure 4, highlighting the difference in the solution structure from the crystal structure conformation shown in Fig. S1.

7. The range of force-field combinations tested for the three proteins in Figure 3 is quite impressive. However, the simulation timescales are only of the order of 230 ns! As was recently shown in a similar benchmark study of a partially disordered protein (10.1021/acs.jctc.3c00148), Rg may fluctuate at the time scales of microseconds. The study also found some force field combinations to cause partial unfolding of initially structured domains, also at the time scale of microseconds. The authors are asked to place their work in the context of the above study and also to comment on the expected behavior of their simulation systems at the longer time scales.

8. To note, the OPC force field is recommended by the AMBER community (https://ambermd.org/AmberModels_proteins.php) to be used with the AMBER ff19SB protein force-field and Li-Merz ions. While the authors have used this water model with some other protein force fields, it might be worthwhile to check the performance of the OPC model when used with the recommended force field combination.

Reviewer #3 (Remarks to the Author):

The work carried out by Linse and Hub studied the effect of the hydration shell on the radii of gyration of five proteins using MD simulations and multiple combinations of force fields and water models. The authors quantified the effective hydration shell of protein systems, allowing a direct comparison to consensus SAXS/SANS experimental data from a worldwide community effort, which is a strong component of this work. Overall, the work is sound, and the methods are described in enough detail to allow reproducibility. The findings described here represent an important milestone in understanding the behavior of biomolecules in solution.

That being said, I have a few major and minor concerns about the work that I would like to see addressed or commented on by the authors.

Major concerns:

1 - It is not clear to me the benefit of using "uncommon" FF-water model combinations. Specifically, for the CHARMM FF combinations, the use of cTIP3P is a historically fundamental part of the standard protein force field development protocol, allowing the community to fine-tune partial charges through water interactions and achieve a proper balance between protein-protein and protein-water interactions. Using a different water model can lead to an unbalanced set of forces, possibly leading to poor predictions (as observed in Figure S6 for CHARMM36m-OPC combination or results mentioned in lines 288-290). I feel references to previous work showing the minimum viability of those uncommon combinations are pivotal to sustain the authors' findings for those combinations.

2 - Differences between the predicted and experimental hydration shell sizes vary across FFs, but the maximum difference observed is still on a sub-Angstrom scale. It would be useful for the readers if the authors could expand on the consequences of such differences, whether they are likely to impact the conformational ensemble. Additionally, the authors could also comment on the choice of using short simulations (50 ns) and the impact that a limited conformational ensemble could have on the predicted Rg.

3 - The authors discuss the effects of protein surface charge on the hydration shell and how different FFs are able to accurately capture such phenomena or not. It would be interesting to see the authors' comments (or even results, if possible) about how the lack of explicit electronic polarization could impact the results presented here. Polarizable FFs such as AMOEBA and Drude could be an interesting path of discussion/investigation for this work since they should, in principle, better capture the electronic nature of the protein/water interface and its correlated forces.

Minor comments:

4 - I really liked Figures 1A and 1B.

5 - I suggest replacing the term "free" when discussing simulation/equilibration with "unrestrained" simulation/equilibration.

6 - Although stated in the figure captions, it is somewhat confusing to understand in the main text (lines 110-129, 148-175, 178-232) whether the results being presented/discussed were obtained from restrained or unrestrained simulations. For the readers' benefit, please add a clarification sentence when discussing those results.

Dear Reviewers,

Many thanks for the careful evaluation of our manuscript and for the constructive criticism, which greatly helped us to strengthen our manuscript.

Please find below our point-by-point response in blue font to the reviewers' remarks in black font. In addition, we provide two PDF files `diff.pdf` and `diff_SI.pdf` that highlight the changes we did to the manuscript and SI, respectively.

Sincerely yours,
Johanna Linse
Jochen Hub

Reviewer #1 (Remarks to the Author):

This is an interesting paper that compares several force-fields and water models to consensus SAXS and SANS data. The paper is well-written and provides an important contribution for the interpretation of SAXS/SANS data and for the assessment of force-field accuracy against scattering data.

My only comment on this work is that a quantity that has been suggested to be very important, the excess water density at the protein surface, is mentioned but never computed in this work. I recommend that the authors to compute this quantity (see e.g. figure 3 of ref. 18) and report how much it varies between force-fields and how much of this variation can explain variations in the results obtained here for different proteins/force-fields.

Thank you for this suggestion. We have added a new Supplementary Figure S7 that shows density profiles of solvent as function of distance from the VdW surface of xylanase for three combinations of protein and water force fields. As expected, the density profiles reveal differences that correlate with the effects of the hydration shell on R_g .

We added to the main text:

"Indeed, the variations of ΔR^{SAS} correlate with variations of the solvent density around the protein surface as shown for three example force field combinations in Fig. S7."

In addition, to further illustrate the hydration shell, we added a Supplementary Movie 1.

Reviewer #2 (Remarks to the Author):

In this manuscript by Linse et. al., the authors examine the effect of molecular dynamics force fields on the simulated hydration shells of five proteins by comparison against small angle scattering data (x-ray and neutron). They found that the hydration shell is dependent on the shape and the charge of the protein and is accurately described by only a few force-field combinations. Overall, the manuscript presents a meticulous detail of radius of gyration calculations for several systems while describing the best practices for comparing simulations with SAS experimental data and provides a very efficient dataset for further improvement of the accuracy of molecular simulations. The work will be a great resource for the molecular modeling community when published.

Addressing the following minor comment will improve the manuscript.

1. For Figure 1, it would be beneficial to have an image of the system with the complete water box for visual clarity, to show how much of bulk water is present in the MD simulation outside the hydration shells as shown in panels A and B.

We agree that an image of a complete simulation system is useful. Since Figure 1 is already quite busy, we now added a simulation system example of xylanase as new Figure S1.

2. The authors describe the presence of three hydration shells in xylanase, Fig 1, using solvent density averaged over the protein surface. However, there is no section in the Methods that describes this averaging procedure, detailing how the surface was determined for carrying out this analysis.

Thank you for pointing this out, as this was indeed missing. We added:

“Density profiles (Fig. 1C) were computed from heavy atom-restrained simulations using gmx genenv module with option -od of GROMACS-SWAXS. The tool builds a series of envelopes with increasing distances from the Van-der-Waals surface of the protein and computes electron density in between pairs of envelopes.”

3. Figure S3 or its variant should be moved to the main text. The RMSD is the most widely used metrics of protein force field and hence will be of direct interest to the readers.

We agree that the RMSD is widely used to test the conformational stability of proteins during MD simulations. However, the purpose of this paper is not to test the quality of force fields. Instead, we merely aim to shed some light on the question why ΔR_g (the increase of R_g owing to the hydration shell) is larger in unrestrained simulations as compared to restrained simulations. The finding is that, in presence of protein flexibility, the water may more favorably adapt to the protein surface, leading to a tighter hydration shell.

We feel that this result is rather a detail and not critical for the key findings of the paper. Thus, for the sake of focus, we prefer keeping the figure in the SI.

4. The time blocks selected for averaging ΔR_g values and calculating standard errors are different for the restrained and free simulations (4.5 ns vs 20 ns) for the same analysis. Were the pure solvent simulations binned into the same time blocks? It is not very clear why they have selected different time blocks for ensuring uncorrelated data?

For the block analysis, it is only important to use statistically independent (uncorrelated) time blocks. In restrained simulation, autocorrelation times are by far shorter as compared to unrestrained simulations. We added:

“Here, for restrained simulations, shorter simulations and time blocks used because restrained simulations exhibit shorter autocorrelation times as compared to unrestrained simulations.”

Information on the binning protocol for the pure-solvent simulation was indeed missing. We now write more precisely:

“Likewise, the pure-solvent simulations were binned into independent time blocks of 4.5 ns.”

5. The authors have done a throughout job of detailing all the average Rg data for the simulations carried out in main text figures and SI figures and tables. To supplement this, it would be useful to add the individual traces of RgProt, to check if there are unusual structural deformations that lie outside the calculated average values, similar to the RMSD data shown in Fig. S3A. It should also help to single out the unusual conformations in the CHARMM36m-OPC simulation that the authors report in Fig. S6 from visual inspection of the corresponding trajectory.

Thank you for this suggestion. In fact, prior to the original submission, we have carefully evaluated RMSD and Rg values (as well as secondary structure changes vs. time) of the CHARMM36m-OPC simulation, and we compared them with other simulations. However, the changes are highly subtle and fluctuating, so they time traces are hardly useful for the reader. Thus, we decided that a more qualitative description of the peculiarities of the CHARMM36m-OPC simulation, together with Fig. a graphical image (Fig. S7), turned out to provide the most useful information for the reader.

6. Following onto the previous comment, it would also useful to visualize the protein conformations that show the best agreement with experimental SAS data. A representative protein conformation along with the hydration shell can be shown for the best performing force-field combinations shown in Figure 4, highlighting the difference in the solution structure from the crystal structure conformation shown in Fig. S1.

All our SAXS curves and Rg values represent ensemble averages involving a solvent fluctuations and conformational dynamics of the protein (to the extent possible within simulation time and to the extent allowed by possible restraints). Thus, selecting a single “best matching” structure would be misleading.

In addition, note that the aim of this study is not to validate protein conformations in MD simulations against experimental SAXS curves, as we have done previously (Chen and Hub, Biophys J 2014). Such comparison would be sensitive to minor random drifts of the protein structure, as common during simulation on the 100s of nanosecond time scale. Instead, we focus only on the **increase** of Rg owing the hydration shell, which hardly depend on minor drift of the protein conformation in unrestrained simulations.

7. The range of force-field combinations tested for the three proteins in Figure 3 is quite impressive. However, the simulation timescales are only of the order of 230 ns! As was recently shown in a similar benchmark study of a partially disordered protein (10.1021/acs.jctc.3c00148), Rg may fluctuate at the time scales of microseconds. The study also found some force field combinations to cause partial unfolding of initially structured domains, also at the time scale of microseconds. The authors are asked to place their work in the context of the above study and also to comment on the expected behavior of their simulation systems at the longer time scales.

The senior authors of the SAXS benchmark study (Trewhella, Vachette et al, Acta Cryst D, 2022, Ref. 26) selected on purpose a set of proteins that are considered as conformationally highly

stable, to avoid that minor changes of experimental conditions at different beamlines lead to different protein ensembles. Specifically, intrinsically disordered proteins (IDPs) as discussed by Sarthak et al. (10.1021/acs.jctc.3c00148) were not considered. We agree that the hydration shell of IDPs is an exciting topic as well, but this is outside the scope of our work. Force fields for IDPs have been discussed by other authors in detail (e.g. Rauscher et al JCTC 2015). To make this more clear, we now refer in the Introduction to “globular proteins” and we added:

“The five proteins considered in the round-robin study were selected to be relatively rigid to avoid complications owing to pronounced flexibility.”

To the methods, we added:

“Since the five proteins considered in the round-robin SAS study are relatively rigid, 230 ns were likely sufficient to sample the most relevant conformation space.”

8. To note, the OPC force field is recommended by the AMBER community (https://ambermd.org/AmberModels_proteins.php <https://ambermd.org/AmberModels_proteins.php>) to be used with the AMBER ff19SB protein force-field and Li-Merz ions. While the authors have used this water model with some other protein force fields, it might be worthwhile to check the performance of the OPC model when used with the recommended force field combination.

Thank you for this suggestion. In response to the reviewer, we now tried to setup simulation with Amber19SB. However, for setting up simulations for GROMACS, Amber19SB is so far only available via Charmm-GUI and not via command line options (because so-called Residue Topology Files (rtp) files for GROMACS have not yet been released for Amber19SB). Consequently, the setup protocol would differ, and we cannot guarantee identical starting conditions. Since we here compare small variations in R_g , we feel that a clearly defined setup protocol is crucial. Hence, we decided to leave Amber19SB with OPC to a future study.

Reviewer #3 (Remarks to the Author):

The work carried out by Linse and Hub studied the effect of the hydration shell on the radii of gyration of five proteins using MD simulations and multiple combinations of force fields and water models. The authors quantified the effective hydration shell of protein systems, allowing a direct comparison to consensus SAXS/SANS experimental data from a worldwide community effort, which is a strong component of this work. Overall, the work is sound, and the methods are described in enough detail to allow reproducibility. The findings described here represent an important milestone in understanding the behavior of biomolecules in solution.

That being said, I have a few major and minor concerns about the work that I would like to see addressed or commented on by the authors.

Major concerns:

1 - It is not clear to me the benefit of using "uncommon" FF-water model combinations. Specifically, for the CHARMM FF combinations, the use of cTIP3P is a historically fundamental part of the standard protein force field development protocol, allowing the community to fine-tune partial charges through water interactions and achieve a proper balance between

protein-protein and protein-water interactions. Using a different water model can lead to an unbalanced set of forces, possibly leading to poor predictions (as observed in Figure S6 for CHARMM36m-OPC combination or results mentioned in lines 288-290). I feel references to previous work showing the minimum viability of those uncommon combinations are pivotal to sustain the authors' findings for those combinations.

Thank you for raising this point. We fully agree that the “uncommon” FF-water model combinations are typically not recommended, although the choice of a most suitable water model may be complicated and depending on the system or observable under consideration. For instance, the standard TIP3P has also been reported to be superior to cTIP3P in the context of peptide folding (<http://doi.org/10.1021/acs.jpcc.6b01316>). Charmm36m has also been reported to work reasonably well with OPC (<https://doi.org/10.1016/j.bpj.2022.11.2282>).

For the present study, however, our aim is for sure not to advertise uncommon FF-water model combinations. Instead, we originally anticipated that the hydration shell contrast would be strongly dominated by the water model, and to a much lower degree by the protein force field. To dissect the role of protein and water model, we decided to simulate with various (including uncommon) FF combinations, which provided the surprising result that *not* the water model, but the protein force field mostly determines the hydration shell contrast. We feel that this represents an important result that will guide developers in further force field refinements. Therefore, we now write more clearly:

“Such uncommon combinations are generally not recommended because protein force fields have been parametrized with respect to specific water models; in this study, however, we considered such uncommon combinations with the aim to dissect effects of the protein force field and of the water model on the hydration shell.”

2 - Differences between the predicted and experimental hydration shell sizes vary across FFs, but the maximum difference observed is still on a sub-Angstrom scale. It would be useful for the readers if the authors could expand on the consequences of such differences, whether they are likely to impact the conformational ensemble. Additionally, the authors could also comment on the choice of using short simulations (50 ns) and the impact that a limited conformational ensemble could have on the predicted R_g .

We used 50 ns only for simulation with restraints on backbone or heavy atoms, thus enabling only side chain fluctuations (if any). Since side chains fluctuate rapidly, 50 ns should be by far sufficient to sample the relevant conformational space. Unrestrained simulations were simulated for 230 ns.

We now write:

“Systems with restraints on heavy atoms or on the backbone were simulated for 50 ns, which is sufficient to sample fluctuations of water molecules or amino acid side chains (if present). Unrestrained simulations were carried out for 230 ns. Since the five proteins considered in the round-robin SAS study are relatively rigid, 230 ns were likely sufficient to sample the most relevant conformation space used to compute the increase of R_g owing to the hydration shell.”

3 - The authors discuss the effects of protein surface charge on the hydration shell and how different FFs are able to accurately capture such phenomena or not. It would be interesting to

see the authors' comments (or even results, if possible) about how the lack of explicit electronic polarization could impact the results presented here. Polarizable FFs such as AMOEBA and Drude could be an interesting path of discussion/investigation for this work since they should, in principle, better capture the electronic nature of the protein/water interface and its correlated forces.

We fully agree that polarizable force fields would be interesting to consider as well. However, considering that the AMOEBA is not supported by GROMACS, while Drude simulations are only available via Charmm-GUI, we would be forced to use different setup protocols, which would require additional technical controls. Thus, we decided to leave polarizable force fields to a future study.

Minor comments:

4 - I really liked Figures 1A and 1B.

Thank you for the encouraging comment.

5 - I suggest replacing the term "free" when discussing simulation/equilibration with "unrestrained" simulation/equilibration.

We agree, we changed the term throughout the manuscript and SI.

6 - Although stated in the figure captions, it is somewhat confusing to understand in the main text (lines 110-129, 148-175, 178-232) whether the results being presented/discussed were obtained from restrained or unrestrained simulations. For the readers' benefit, please add a clarification sentence when discussing those results.

We agree. We added a few explicit statements to the text to emphasize whether restrained or unrestrained simulations are discussed.

Reviewers' comments:

Reviewer #1 (Remarks to the Author):

In the original revision, I asked to compute the excess solvent density as in figure 3 of ref.18. where the difference between the bulk water RDF and the RDF with respect to the protein atoms is computed. By subtracting the bulk water RDF, information on the specific effect of the protein in inducing perturbations in the water structure can be obtained.

The authors apparently just plotted in Fig. S7 the water density with respect to the protein atoms, modulo a constant density. This quantity is less interesting because it is affected by intrinsic differences in the bulk RDF of different water models. I urge the authors to subtract from the plots in figure S7 the RDF of bulk water, as in ref. 18. The resulting plots should be more informative about the extent of perturbation that the protein induces in the water layer using different water models.

Reviewer #2 (Remarks to the Author):

Unfortunately, the authors have not adequately addressed most of the comments from the previous round of review, including the very basic task of performing an additional MD simulation using an appropriate combination of protein and water models.

However, this reviewer would not have very strong objections if the editor found the paper suitable for publication in Communications Chemistry.

Reviewer #3 (Remarks to the Author):

The authors addressed the questions I had and the changes they proposed improve the overall interpretation of the results presented here. Thus, I suggest the acceptance of this paper.

Reviewer #1 (Remarks to the Author):

In the original revision, I asked to compute the excess solvent density as in figure 3 of ref.18. where the difference between the bulk water RDF and the RDF with respect to the protein atoms is computed. By subtracting the bulk water RDF, information on the specific effect of the protein in inducing perturbations in the water structure can be obtained. The authors apparently just plotted in Fig. S7 the water density with respect to the protein atoms, modulo a constant density. This quantity is less interesting because it is affected by intrinsic differences in the bulk RDF of different water models. I urge the authors to subtract from the plots in figure S7 the RDF of bulk water, as in ref. 18. The resulting plots should be more informative about the extent of perturbation that the protein induces in the water layer using different water models.

Reply: Thank you for describing this suggestion in more detail, since we indeed misunderstood your point during the first round of revision.

We agree that the difference of the solvent density profiles (or RDFs) between (i) the protein hydration shell relative to (ii) bulk water is an important quantity, since this difference manifests in the modified R_g values. Thus, following the reviewer's suggestion we have now computed the solvent density profile around a volume of restrained bulk water (instead of a restrained protein), where the volume of restrained water takes approximately the volume of xylanase.

In qualitative agreement with the analysis by Merzel and Smith (PNAS 2002, Ref. 18), we find that the water is by far more structured at the protein surface as compared to water around a volume of bulk water (see updated Figure 1C, dashed line). Notably, the difference between protein and bulk water is somewhat more pronounced as compared to the effects reported by Merzel and Smith, possibly because we carried out the analysis with restraints on all heavy protein atoms.

We present the new analysis in Figure 1C (dashed line), a new Figure S8 and a new Supplementary Methods (see SI pages 8-9). In the main text we now write:

Page 4:

"To test whether solvent density modulations owing to water-protein interactions differ from density modulation owing to the internal structure of bulk water, we carried out a simulation of bulk TIP4P/2005s water, in which water molecules within a volume similar to the volume of xylanase were restrained (Supplementary Methods). Density modulations around such restrained bulk water are by far smaller as compared to the density modulations at the protein surface (Fig. 1C, compare dashed dark green with solid magenta line), in line with previous reports.¹⁸"

Page 10:

"To test whether ΔR^{SAS} variations among force fields are explained by variations of the hydration shell densities, we computed solvent density profiles as function of distance R from the Van-der-Waals surface of xylanase using the force field combinations

CHARMM36m-cTIP3P, ff15fb-TIP3P-FB, or ff99SBws-TIP4P/2005s (Fig. S7), which led to small, medium, or large ΔR^{SAS} values (Fig. 3D, blue, green, and black bar, respectively). In addition, density profiles around xylanse were compared with density profiles around volumes of restrained bulk water modeled with cTIP3P, TIP3P-FB, or TIP4P/2005s (Fig. S8), thereby comparing the hydration shell structures with the structure of bulk water.¹⁸ We find that the height and width of the first solvent density peak at $R \approx 0.85 \text{ \AA}$ correlate with ΔR^{SAS} values, both if taken from the total solvent density profiles (Fig. S7) or if taken from the density profile relative to the bulk water structure (Fig. S8B/D). This analysis suggests that modulations of the hydration shell structure among different force fields indeed manifest in variations of ΔR^{SAS} .

We furthermore renamed the previous “Supplementary Methods” to “Supplementary Results” for clarity.

REVIEWERS' COMMENTS:

Reviewer #1 (Remarks to the Author):

The authors have now addressed my concern, I believe that the paper is suitable for publication and will be of great interest for both the computational and experimental community.